# Thermal Cycling Stimulation via Nasal Inhalation Attenuates Aβ_25–35_-Induced Cognitive Deficits in C57BL/6 Mice

**DOI:** 10.3390/ijms262010236

**Published:** 2025-10-21

**Authors:** Guan-Bo Lin, Hsu-Hsiang Liu, Yu-Yi Kuo, You-Ming Chen, Fang-Tzu Hsu, Yu-Wei Wang, Yi Kung, Chien Ching, Chih-Yu Chao

**Affiliations:** 1Laboratory for Medical Physics & Biomedical Engineering, Department of Physics, National Taiwan University, Taipei 106319, Taiwan; gblin@phys.ntu.edu.tw (G.-B.L.); yykuo@phys.ntu.edu.tw (Y.-Y.K.); tsyr8924503@phys.ntu.edu.tw (F.-T.H.); a13956@phys.ntu.edu.tw (Y.K.); 2Molecular Imaging Center, National Taiwan University College of Medicine, Taipei 100233, Taiwan; hhliu@phys.ntu.edu.tw (H.-H.L.); ymchen@phys.ntu.edu.tw (Y.-M.C.); f10245021@phys.ntu.edu.tw (Y.-W.W.); albertj8808@phys.ntu.edu.tw (C.C.); 3Biophysics Division, Graduate Institute of Applied Physics, National Taiwan University, Taipei 106319, Taiwan

**Keywords:** Alzheimer’s disease, β-amyloid, hyperthermia, nasal inhalation, mild stress

## Abstract

Alzheimer’s disease (AD) remains a significant public health challenge, with current treatments limited partly due to the difficulty of delivering therapeutics across the blood–brain barrier (BBB). The nose-to-brain (N-2-B) pathway offers a promising alternative to circumvent the BBB, but no drugs have yet been clinically applied via this route for AD. Mild stress is thought to activate intrinsic protective mechanisms against neurodegeneration, but traditional methods lack specificity and practicality. To address this, we propose the inhalation of mildly heated air as thermal stimulation, which utilizes the N-2-B pathway to induce mild stress and stimulate cerebral activity. This study employs thermal cycling-hyperthermia (TC-HT) in developing thermal cycling-stimulation via nasal inhalation (TCSNI), providing cyclic stimulation to maintain pathway activity while minimizing thermal injury. In C57BL/6 mice, TCSNI showed no adverse olfactory effects. In β-amyloid (Aβ)-treated mice, TCSNI significantly enhanced cognitive performance in Y-maze and novel object recognition (NOR) assessments, suggesting cognitive improvement. Mice hippocampal protein analyses indicated a reduction in Aβ accumulation, alongside increased expression of heat shock protein 70 (HSP70), insulin-degrading enzyme (IDE), and phosphorylated Akt (p-Akt). These results suggest that N-2-B-delivered TCSNI effectively modulates protein expression and enhances cognitive function, highlighting its potential for further exploration in AD treatment.

## 1. Introduction

Alzheimer’s disease (AD) is a leading cause of dementia and represents an escalating threat to public health [1]. The abnormal accumulation of β-amyloid (Aβ) in the brain is believed to play a significant role in the oxidative stress and damage that contribute to the pathogenesis of AD [2,3,4]. Furthermore, Aβ aggregation is linked to synaptic dysfunction and neuronal loss, which are critical factors in cognitive decline among AD patients [5,6]. Consequently, Aβ has become a primary target for therapeutic interventions in AD [5,6,7]. Recently, drugs developed based on the Aβ hypothesis, such as aducanumab and lecanemab, have received FDA approval [8,9]. However, these drug treatments are expensive and carry risks of adverse effects, including brain edema and abnormal bleeding [10,11], which complicate their broader implementation. While pharmacotherapy remains the predominant treatment strategy, the blood–brain barrier (BBB) constrains advancements in new drug development and application [12,13], highlighting the need for further exploration of alternative therapies. The integration of mild stress-based non-pharmacological interventions into healthcare strategies offers a promising opportunity for enhancing health outcomes [14,15,16,17,18]. The design of optimal and well-controlled mild stress stimuli has garnered considerable interest due to their potential to stimulate neuronal cells, improve cognitive function, and possibly extend lifespan, thereby offering beneficial effects in the context of neurodegenerative diseases [19,20,21]. These benefits are primarily mediated through the principle of hormesis, wherein exposure to low levels of stressors triggers adaptive cellular responses that enhance resilience against neurodegenerative processes [22,23,24,25]. Research has indicated that mild stress may upregulate heat shock proteins (HSPs), which are associated with decreased Aβ aggregation [26,27]. Exercise and sauna use are widely recognized as physiological stimuli capable of inducing mild stress in humans [19,20,21,28,29]. However, despite their general health benefits, these long-term approaches are time-consuming and lack definitive evidence of curative effects [30,31,32,33]. Therefore, the exploration of novel and convenient alternative approaches is warranted.

The nose-to-brain (N-2-B) pathway presents a promising alternative approach for drug delivery by bypassing the BBB [34,35]. Various pharmacological agents, including donepezil and rivastigmine, as well as natural compounds such as quercetin and curcumin, have been investigated in relation to the N-2-B pathway [36,37]. Moreover, the inhalation of insulin has attracted considerable interest in studies pertaining to this pathway [36,38]. Concurrently, aromatherapy is recognized as a potentially effective complementary or alternative therapy for dementia treatment [39,40,41], as inhaled compounds may mitigate symptoms through activation of olfactory pathways and modulation of emotion- and cognition-related brain regions [42,43,44]. However, despite extensive investigations into numerous drugs and bioactive compounds, no medication has yet been successfully administered via this route for the clinical treatment of AD [36], indicating a need for further exploration of the N-2-B pathway.

While pharmacological treatments and complementary therapies such as aromatherapy have demonstrated some progress, their overall efficacy remains limited. Building on the concept of olfactory stimulation and extending beyond traditional pharmacological approaches, we propose, for the first time, the innovative application of inhaling mildly heated air as a form of thermal stimulation via the N-2-B pathway to activate the brain and induce mild stress. Thermal stimulation has long been recognized for its potential benefits across various medical disciplines [45,46,47]. Appropriate hyperthermia (HT) has been shown to enhance the expression of heat shock proteins (HSPs), particularly heat shock protein 70 (HSP70), which is considered advantageous for improving AD [48,49,50]. Additionally, studies indicate that HT can elevate the expression of insulin-degrading enzyme (IDE), which is thought to mitigate Aβ accumulation [51,52]. Furthermore, thermal stress has been demonstrated to activate Akt signaling pathways [53,54], and the neuroprotective effects of phosphorylated Akt (p-Akt) in AD have been extensively discussed [54,55,56,57]. Recently, sauna use has gained attention as a thermal stimulation practice for dementia prevention [30,31,48]; however, while associated with potential cognitive benefits, its therapeutic efficacy remains inconclusive [30,31]. Moreover, accessibility issues and safety concerns, particularly for elderly individuals with cardiovascular risk factors, may limit its practicality [58,59].

In this study, we aimed to explore the potential effects of inhaling slightly heated air as a novel method of mild thermal stimulation via nasal inhalation. Compared with systemic heating, which involves the entire body, our approach uses localized nasal inhalation. To mitigate the risks of continuous heat exposure, we adapted our thermal cycling-hyperthermia (TC-HT) technique into thermal cycling-stimulation via nasal inhalation (TCSNI), which delivers repeated cycles of heated and ambient air to reduce stress while sustaining HSP-related signaling. We conducted a comparative analysis of the effects of TCSNI and continuous hyperthermia stimulation via nasal inhalation (HTSNI), with a particular emphasis on their respective influences on olfactory function in C57BL/6 mice. Additionally, we assessed the capacity of TCSNI to enhance cognitive performance, utilizing the Y-maze and novel object recognition (NOR) tests [60,61,62,63] in a mouse model exhibiting cognitive impairment induced by intracerebroventricular (i.c.v.) injection of Aβ_25–35_ [64]. Furthermore, we investigated the expression of heat- and stress-responsive proteins associated with neurodegeneration, including HSP70, IDE, and phosphorylated Akt (p-Akt). These proteins are known to play a role in antioxidant defense, promote cell survival, and facilitate the clearance of Aβ accumulation [48,49,50,51,52,53,54,55,56,57]. This study aims to contribute new insights into the therapeutic potential of nasal inhalation-based thermal stimulation and its implications for the prevention or mitigation of cognitive decline, thereby establishing a foundation for future applications in neurodegenerative diseases (NDDs).

## 2. Results

### 2.1. Effects of HTSNI and TCSNI on Olfactory Function in Mice

The objective of this research is to explore a novel hyperthermic stimulation technique administered through the nasal cavity. Prior to conducting cognitive assessments in AD mice, we first evaluated the safety of hyperthermic stimulation approaches, specifically HTSNI and TCSNI, by monitoring potential adverse effects, as illustrated in Figure 1. Both HTSNI and TCSNI are methods of hyperthermic stimulation targeting the nasal cavity, necessitating an examination of the olfactory function in mice after treatments. The buried food test and odor sensitivity test are common methods for evaluating the olfactory capabilities of mice [65,66]. Our experimental results indicated that following TCSNI treatment, the latency to find food (14.0 ± 3.5 s) did not significantly differ from that of the control group (17.5 ± 2.8 s) in the buried food test (Figure 1A). Conversely, the HTSNI group exhibited a significantly prolonged latency (70.1 ± 4.0 s) in comparison to both the control and TCSNI groups. Additionally, in the odor sensitivity assessment, the time spent sniffing odor stimuli did not differ significantly between the control and TCSNI groups across the range of odor dilutions. Notably, at a dilution of 10^−3^, the HTSNI group exhibited significantly shorter sniffing time (1.4 ± 0.3 s) compared to the control (3.5 ± 0.5 s) and TCSNI (3.0 ± 0.4 s) groups. Furthermore, at a dilution of 10^−2^, mice in the HTSNI group spent significantly less time sniffing (1.2 ± 0.3 s) than both the control (4.2 ± 0.5 s) and TCSNI (4.3 ± 0.5 s) groups, as shown in Figure 1B. These results suggest that the mice subjected to the HTSNI exhibited diminished olfactory sensitivity, indicating a reduced capacity to detect and respond to odor cues at concentrations that remained effective for the control and TCSNI groups. In view of this, we applied TCSNI in the subsequent evaluations to determine its potential to enhance recognition in the AD mouse model.

### 2.2. TCSNI Attenuates Aβ-Induced Cognitive Impairments in Mice

Given that memory impairment is a defining characteristic of AD [67], research utilizing mouse models commonly incorporates Y-maze and NOR tests to assess cognitive deficits [60,61,62,63]. Spontaneous alternation in the Y-maze serves as a well-established indicator of short-term spatial working memory [60,61], while the NOR test is a widely recognized paradigm for evaluating both short-term and long-term recognition memory [62,63]. In this study, we employed these assessments to analyze the effects of TCSNI on Aβ-injected mice. As illustrated in Figure 2A, there was no significant difference in the number of arm entries in the Y-maze test across all groups, indicating that i.c.v. injection of phosphate-buffered saline (PBS) or Aβ, along with TCSNI treatment, did not affect the locomotor activity of the mice. Furthermore, the spontaneous alternation index was significantly diminished in the Aβ group compared to the control group, decreasing from 64.8 ± 2.1% to 50.4 ± 1.8%. Importantly, our TCSNI treatment resulted in an increase in the spontaneous alternation index, suggesting a potential amelioration of Aβ-induced deficits in short-term spatial working memory, as demonstrated in Figure 2B. In addition, the results from the short-term and long-term NOR tests provide compelling evidence for the efficacy of TCSNI in mitigating recognition memory deficits induced by Aβ. As shown in Figure 2C, the discrimination index in the short-term NOR tests exhibited a notable reduction in the Aβ group relative to the control group, decreasing from 63.7 ± 4.5% to 10.5 ± 7.9%. Notably, it was found that our TCSNI treatment resulted in a significant improvement in the discrimination index, which was recorded at 56.3 ± 4.9% following the TCSNI treatment. Additionally, the long-term NOR test revealed a more pronounced memory impairment due to Aβ, with the discrimination index declining from 48.2 ± 3.1% to 4.3 ± 7.0%. Following TCSNI treatment, the results revealed that the discrimination index was restored to 43.4 ± 6.3%, as demonstrated in Figure 2D. Collectively, these findings suggest that TCSNI treatment effectively alleviates Aβ-induced cognitive deficits, as evidenced by the results from both the Y-maze and NOR tests.

### 2.3. TCSNI Down-Regulates Aβ Accumulation and Elevates HSP70, IDE, and p-Akt Levels in the Mouse Hippocampus

Given that Aβ is a primary target in AD therapeutic strategies [5,6,7], this study initially assessed its accumulation in the hippocampus of mice. As shown in Figure 3A, the Aβ-injected group exhibited a 1.50-fold increase in Aβ levels compared to the control group. Importantly, TCSNI treatment significantly reduced the hippocampal Aβ levels, decreasing them from 1.50-fold to 0.90-fold of the control group, thereby demonstrating a substantial inhibitory effect. Furthermore, HSP70, a critical stress response protein that acts as a molecular chaperone to prevent protein misfolding, has been implicated in the modulation of the Aβ aggregation process [68,69,70]. In vivo studies have indicated that Aβ administration may lead to a slight increase in HSP70 levels, although this did not reach statistical significance [71,72]. In the present study, HSP70 levels were also evaluated, revealing that the hippocampal HSP70 levels in the Aβ-injected group were not significantly different from those in the control group. Notably, it was found that TCSNI treatment resulted in a significant upregulation of HSP70 levels, reaching 1.61-fold of the control group, as shown in Figure 3B. Additionally, IDE has long been recognized for its role in Aβ clearance, with alterations in its activity or expression potentially contributing to Aβ accumulation and an increased AD risk [51,73,74]. As shown in Figure 3C, Aβ injection in mice led to a reduction in hippocampal IDE levels to 0.74-fold of the control. This study found that TCSNI treatment effectively reversed this decline, significantly elevating IDE expression to 1.44-fold of the control group. Moreover, Akt is known to play a crucial role in neuronal survival signaling [75,76], and a reduction in Akt activation has been documented in AD models [55,56,57,77,78]. Given that Akt is responsive to stress and thermal stimuli [53,54,79], it warranted further investigation in this study. As illustrated in Figure 3D, Aβ injection in mice resulted in a significant downregulation of the p-Akt/t-Akt ratio (0.83-fold), which was used to accurately represent the phosphorylation level relative to total Akt expression, while TCSNI treatment significantly enhanced this ratio to 1.57-fold compared to the control. Overall, these findings suggest that TCSNI-induced Aβ removal may be associated with the upregulation of HSP70, IDE, and the p-Akt/t-Akt ratio in the hippocampi of the examined mice.

## 3. Discussion

AD remains a significant medical challenge globally. The BBB presents a considerable impediment to the targeted delivery of drugs for the treatment of AD [12,13]. In contrast, the N-2-B pathway has emerged as a promising alternative for circumventing the BBB, with research indicating its potential therapeutic benefits for brain disorders and brain cancer [34,35,36]. Additionally, aromatherapy is recognized as a complementary or alternative therapeutic approach that may offer advantages in the treatment of dementia [41]. Despite the increasing interest in the N-2-B pathway as a means to bypass the BBB, no pharmacological agents have yet received clinical approval for AD treatment via this route [36]. In light of the absence of a definitive cure for AD and the limited effectiveness of existing pharmacological treatments, there has been a shift towards exploring alternative strategies [80,81,82].

The literature suggests that mild stress stimuli can enhance the functional capacity of organisms and yield beneficial biological effects on cellular processes [20,83,84,85]. Moreover, non-pharmacological interventions, such as exercise and sauna use, which induce mild stress, are believed to promote human health, potentially extending lifespan and improving cognitive function [28,29,84,85]. Among these interventions, whole-body heating through sauna has garnered significant attention, and sauna-like conditions have been evaluated [28,30,31,86]. However, due to limitations such as time consumption, practical challenges, and the lack of curative effects, it is imperative to investigate more effective and safe methods for inducing the desired mild stress.

In this study, we propose a novel methodology using the inhalation of mildly heated air as a thermal stimulus to induce mild stress responses. This technique diverges from traditional whole-body heating methods by specifically targeting the nasal cavity with warm air. However, it is imperative to assess the safety of this approach prior to conducting further experimental trials. Previous research has demonstrated that mice can experience thermal collapse when subjected to extreme heat conditions [87]. In contrast to the aforementioned scenario, which involves the exposure of the entire body to elevated temperatures, our study focuses on localized nasal heating through gas delivery. Additionally, the TC-HT method, developed in our earlier studies, has shown efficacy in both in vitro and in vivo models with minimal adverse effects, primarily due to its use of intermittent stimulation intervals that help to mitigate excessive damage [52,54]. Consequently, we assessed the olfactory function in mice subjected to HTSNI and TCSNI. As illustrated in Figure 1, mice receiving HTSNI exhibited impaired olfactory function, as evaluated through the buried food test and the olfactory sensitivity test, whereas the TCSNI-treated group maintained olfactory function comparable to that of the untreated control group. These findings suggest that TCSNI is a safer approach compared to HTSNI.

Memory impairment is a hallmark of AD [67]. Research utilizing mouse models often employs the Y-maze and NOR tests to assess memory deficits [60,61,62,63]. Considering that TCSNI preserves olfactory function in mice, it was selected as the thermal stimulation method for behavioral assessments in mice with cognitive impairments, thereby minimizing potential sensory disturbances, particularly in light of the olfactory decline observed with HTSNI. This study explored the ameliorative effects of TCSNI on Aβ-induced memory impairment and demonstrated that TCSNI significantly improved the spontaneous alternation index and the ability to recognize novel objects, as shown in Figure 2. These results indicate that TCSNI has considerable potential for improving memory deficits and cognitive dysfunction in vivo in a safe manner.

Although traditional continuous heating HT stimulation holds potential to induce beneficial mild stress, existing literature suggests that inappropriate HT may yield no advantages [84,85,88] and could contribute to an increased accumulation of reactive oxygen species (ROS), thereby exacerbating oxidative stress and resulting in cellular damage [88,89,90]. In our previous in vitro study, HT was found to be ineffective in reducing ROS and exhibited less protective effects compared to TC-HT against cytotoxicity induced by hydrogen peroxide and Aβ [54]. Furthermore, another study utilizing thermofoil indicated that continuous HT application was less effective than TC-HT in ameliorating memory impairment in vivo [52]. These findings suggest that utilizing cyclical thermal stimulation instead of continuous heating may provoke distinct biological responses [52,54], potentially linked to the activation and saturation of stimulated protein pathways. For example, in the domain of olfactory research, prolonged and continuous exposure to specific stimuli can result in olfactory adaptation or receptor desensitization [91,92,93,94], which may influence the efficacy of aromatherapy interventions. Consequently, the integration of diverse scents or the implementation of intermittent scent exposure is deemed a critical strategy to maintain receptor activation and therapeutic effectiveness [95,96]. Similarly, in thermal stimulation studies, the phenomenon of thermal adaptation has been examined and identified as a significant factor contributing to the diminished effectiveness of traditional HT treatments [97,98]. In contrast to HT, the TC-HT technique employs periodic temperature variations, which may help prevent signaling desensitization and better sustain heat-induced protein responses. Furthermore, TC-HT methods are characterized by an improved cellular safety profile, potentially augmenting both the efficacy and practicality of thermal stimulation, thereby positioning them as a preferable alternative to conventional HT approaches [52,54]. In this study, while HTSNI was observed to impair olfactory function, the application of TCSNI did not exhibit such adverse effects and was found to improve memory and cognition deficits (Figure 1 and Figure 2).

To investigate the molecular mechanism, we examined Aβ expression levels in the hippocampus, given that the clearance of toxic Aβ deposits serves as a critical indicator in AD therapy [99,100]. As illustrated in Figure 3A, the application of TCSNI resulted in a significant reduction in Aβ expression levels in the hippocampus. Additionally, we examined several proteins that may play a role in Aβ clearance. Notably, HSP70, a prominent heat shock protein activated by thermal stimuli, has garnered considerable attention for its potential to mitigate the progression of AD [68,69,70]. HSP70 primarily functions to prevent protein misfolding and has also been implicated in influencing the aggregation process of Aβ [68,69,70]. Our findings indicate that the TCSNI application significantly elevated HSP70 expression in the hippocampus of mice, as shown in Figure 3B. Furthermore, HSP70 has been suggested to upregulate the expression of IDE, a well-known Aβ-degrading enzyme, thereby facilitating Aβ clearance [68]. Previous studies have demonstrated that IDE is induced by stress and possesses heat shock-like characteristics [51,52,101]. Moreover, IDE has been found to regulate the levels of extracellular Aβ through proteolytic activity, thereby alleviating its neurotoxic effects [102,103]. Research indicates that diminished IDE activity and expression may correlate with Aβ accumulation and an elevated risk of AD [51,73,74], suggesting that enhancing IDE levels could represent a viable therapeutic strategy for AD [74]. Consistent with these findings, our study reveals that TCSNI treatment significantly restores the Aβ-induced reduction in IDE levels within the hippocampus of mice, as shown in Figure 3C.

In addition to IDE and HSP70, Akt emerged as another key protein of interest in our study. Akt is integral to mediating neuronal survival signaling, growth, and synaptic plasticity, all of which are essential for brain development and function [75,76]. A decline in Akt activation has been observed in AD models, whereas an increase in Akt activation has been linked to protective effects in both in vitro and in vivo models [55,56,57,77,78]. Importantly, Akt is also recognized as a protein responsive to stress and thermal stimulation [53,54,79]. Furthermore, research has indicated that Akt may mediate HSP70 expression, thereby providing neuroprotection [78]. Additionally, it has been suggested that the Akt signaling pathway may ameliorate cognitive impairment in AD by promoting IDE expression and facilitating Aβ degradation [104]. In this study, we detected Akt protein levels in the hippocampus of mice along with HSP70 and IDE proteins. Our results demonstrate that the Aβ-induced decline in p-Akt was restored following TCSNI application, as illustrated in Figure 3D. Collectively, these findings suggest that TCSNI induces mild thermal stress, which may upregulate the expression of HSP70, IDE, and p-Akt in the hippocampus of Aβ-injected mice. This molecular response may contribute to the observed reduction in hippocampal Aβ levels and the cognitive improvements in the treated mice, consistent with the hypothesis of mild stress-induced self-protection.

Collectively, this study introduces a novel periodic hyperthermic approach, TCSNI, administered through nasal inhalation to induce mild stress as a potential therapeutic strategy for AD. The stress response is anticipated to activate the cellular self-defense mechanisms [105], with the effects being contingent upon the intensity and duration of the stressor, which may lead to cellular adaptation or apoptosis [106]. Prior research has demonstrated that preconditioning cells with non-lethal stress can reduce cellular damage and enhance cell survival [54,107]. The present study proposes an effective method for inducing mild stress in vivo with minimal impact on olfactory function through TCSNI, which has been shown to enhance cognitive performance in Aβ-injected mice, as evidenced by Y-maze and NOR tests. Additionally, TCSNI was found to decrease Aβ expression and significantly upregulate proteins associated with Aβ degradation and cell survival, such as HSP70, IDE, and p-Akt, in the hippocampus of mice subjected to i.c.v. Aβ injection. Further research is necessary to elucidate the role of additional stress-induced proteins, particularly those linked to neuroprotection and Aβ metabolism, and to validate these findings across different species and in humans. While our current data primarily reflect some improvements in cognitive behavior, the duration and sustainability of these effects remain to be investigated in future studies. In addition, although HTSNI impaired olfactory function, comparisons between TCSNI and HTSNI, as well as assessments in other brain regions, may reveal distinct characteristics and represent promising directions for future research. Moreover, the potential synergistic effects of TCSNI in conjunction with pharmacological treatments merit further exploration. Given its non-invasive and non-pharmacological nature with a straightforward application concept, TCSNI holds potential for clinical translation in AD patients; however, further studies are required to assess the long-term efficacy and sustained cognitive benefits in clinical settings. Overall, this study provides new insights into the development of alternative therapeutic strategies for AD, underscoring the potential of TCSNI as a novel non-invasive intervention.

## 4. Materials and Methods

### 4.1. Experimental Animals and Housing

C57BL/6 male mice (6–8 weeks old, 22–26 g) were purchased from the National Laboratory Animal Center (Taipei, Taiwan) and housed at the National Taiwan University Animal Resources Center. The mice were provided with ad libitum access to food and water and were maintained at a controlled temperature of 23 ± 2 °C, under a 12:12 h light/dark cycle (lights switched on at 7:00 a.m.). All experimental procedures adhered to the established guidelines of the Animal Ethical Committee of National Taiwan University (approval protocol number: NTU-112-EL-00158). The mice were randomly allocated into two primary groups. The 1st group was subdivided into control, TCSNI, and HT subgroups, while the 2nd group was categorized into control, Aβ, and Aβ + TCSNI subgroups.

### 4.2. β-Amyloid Administration

Aβ_25–35_ (Sigma-Aldrich; Merck KGaA, Darmstadt, Germany) was prepared as a stock solution at a concentration of 1 mg/mL in 0.9% sterile saline solution and was fibril-aggregated at 37 °C for 7 days. On day 0, 10 µL of the aggregated Aβ was i.c.v.-injected into the brains of the mice in the Aβ and Aβ-TCSNI groups, while the mice in the control group received an i.c.v. injection of 10 µL PBS.

### 4.3. HTSNI and TCSNI Applications

This study employed a novel nasal inhalation-based thermal stimulation system. As illustrated in Figure 4A, the system utilized a water bath method to heat 3% isoflurane anesthetic gas and regulated the duration of gas heating and cooling through a timer-controlled solenoid valve. In the HTSNI application, the solenoid valve was fixed to ensure a consistent release of heated air, maintaining a temperature of 47.5 ± 0.5 °C for 30 min. On the other hand, in the application of TCSNI treatment, the timer-controlled solenoid valve was programmed to execute a repetitive 10-cycle process, with each cycle comprising a 3 min heating phase (47.5 ± 0.5 °C) followed by a 1 min cooling phase. The actual air temperature entering the nasal cavity of the mice was measured and monitored using a K-type thermocouple positioned at the end of the pipeline, with the measured temperature results presented in Figure 4B. All experimental groups were subjected to the same isoflurane anesthesia to ensure consistent sedation during the procedures. Prior reports in various AD mouse models suggest that short-term isoflurane exposure is generally well tolerated and does not alter Aβ-related pathology [108,109].

### 4.4. Experimental Design

For the 1st group, as depicted in Figure 5, the respective thermal application was employed in the mice of the HTSNI and TCSNI groups on days 7, 10, and 13. Buried food and odor sensitivity tests were conducted on days 15 and 16, respectively. In the 2nd group, 10 µL Aβ solution was i.c.v.-injected into the brains of mice in the Aβ and Aβ + TCSNI groups on day 0, while mice in the control group received an i.c.v. injection of 10 µL PBS. The TCSNI thermal application was applied to the mice in the various groups on days 7, 10, and 13. The Y-maze test was conducted on day 14, followed by the NOR tests on days 15 and 16. Following the NOR test, the mice were weighed and subsequently euthanized for the collection of brain tissues.

### 4.5. Buried Food Test

On day 14, food pellets were removed from the cages, and the mice underwent an overnight fasting period. The test was performed on day 15, following a 1 h acclimatization period in the testing room. Each mouse was then individually placed into a clean cage containing clean bedding 3 cm in depth, where they were allowed to acclimate for 5 min. This acclimatization aimed to minimize the potential confounding effects of exploring a novel environment during the test. A small food pellet, approximately 5 mm in diameter, was buried beneath 1 cm of bedding in a randomly selected corner of the cage, and the mouse was subsequently introduced into the cage. The site of mouse placement and the location where the small pellet was buried remained consistent throughout the experiment. The latency, defined as the time taken for the mice to retrieve the small pellet using their forepaws, was recorded in seconds with a maximum duration of 15 min.

### 4.6. Odor Sensitivity Test

The odor sensitivity test in mice was conducted on day 16 after 1 h of acclimatization in the testing room. Each mouse was individually placed into a clean cage, which was pre-placed with a dry cotton swab without bedding. The mice were allowed to acclimate to the cage for 30 min to reduce the effects of exploring a novel environment during the test. Following this acclimatization, video recording began and lasted for 3 min, during which the dry cotton swab was replaced with a cotton swab soaked in distilled water. After the initial 3 min, the cotton swab was sequentially replaced with diluted solutions of vanilla extract (Flavorganics, Newark, NJ, USA) at concentrations of 1/100,000, 1/10,000, 1/1000, and 1/100. An increase in sniffing behavior was anticipated as the odor concentration approached the olfactory threshold for the mice. After acquiring all of the olfactory sensitivity and preference videos, a timer was used to time all investigations of the cotton swabs within three min. The investigation time was defined as any period when the mouse’s nares were <1 mm from the cotton swabs. The recorded latency did not include periods when the mouse was not actively investigating, chewing, or licking the swab.

### 4.7. Y-Maze Test

The Y-maze test was employed to evaluate short-term spatial working memory. The apparatus consisted of three identical symmetric arms (40 cm × 13 cm × 6.5 cm) arranged at 120° angles from a central junction. Each mouse, having no prior familiarity with the maze, was individually placed at the center and allowed to freely explore all three arms to assess spontaneous alteration for 8 min. Following each trial, the maze was cleaned with 70% ethanol to eliminate any olfactory cues. Behavioral performance was documented, and the entries into the maze arms were designated as ‘A’, ‘B’, and ‘C’. An arm entry was defined as the complete entry of the mouse’s hind paws into the arm. A correct alternation was characterized by sequential entries into the three distinct arms (i.e., ABC, ACB, BAC, BCA, CAB, and CBA) in overlapping triplet sets. The spontaneous alternation index was calculated using the following formula: (number of correct alternations)/(total number of arm entries − 2) × 100, with the total number of arm entries also serving as an indicator of general locomotor activity.

### 4.8. Novel Object Recognition Test

The NOR test was conducted to evaluate recognition memory in mice during days 14 to 16 of the study. The test was performed in an open field arena (40 cm × 40 cm × 30.5 cm) over three consecutive days. On day 14 (habituation phase), each mouse was allowed to freely explore the empty arena for 8 min to acclimate to the testing environment. On day 15 (familiarization phase and short-term recognition memory test phase), two identical familiar objects (object A) were placed near the diagonal corners of the arena, allowing the mouse to explore the arena for 8 min. Following a 1 h retention interval in the home cage, the mouse was reintroduced to the arena for the short-term recognition memory test, during which one of the original identical familiar objects (object A) was substituted with a novel object (object B). The mouse’s behavioral responses were recorded via an overhead video camera for 8 min. On day 16, to evaluate long-term recognition memory, the mice were again placed in the arena and allowed to explore the same familiar object from the training phase (object A) and a new novel object (object C) for 8 min. This experimental design facilitated the assessment of both short-term and long-term object recognition memory. The familiar objects were identical in shape, material composition, and color, while the novel objects exhibited diverse shapes, sizes, and colors in comparison to the familiar objects. After each assessment in all phases, all olfactory cues were eliminated using 70% ethanol. Object exploration was quantified as the time spent by the mouse sniffing or touching the object with the nose and/or forepaws. The discrimination index was calculated using the formula (Tn − To)/(Tn + To), where Tn and To represent the exploration times for the novel and familiar objects, respectively.

### 4.9. Collection of Brain Tissue and Preparation of Samples for Western Blot Analysis

After NOR tests, the mice were euthanized using carbon dioxide gas, and their brains were extracted following confirmation of death. The brains were rinsed with PBS (HyClone; GE Healthcare Life Sciences, Chicago, IL, USA) supplemented with 1% fresh protease inhibitor cocktail (EMD Millipore, Billerica, MA, USA). The hippocampus tissue was homogenized in 500 µL of ice-cold RIPA lysis buffer (MilliporeSigma) containing phosphatase and protease inhibitors using a Q125 Sonicator (Qsonica, Newton, CT, USA). The homogenates were then centrifuged at 23,000× *g* at 4 °C for 30 min. Protein concentrations of the samples were determined using the Bradford protein assay (cat. no. BRA222; BioShop Canada). The protein samples were subsequently stored at −80 °C for further Western blotting analyses.

### 4.10. Western Blot Analysis

20 µg of homogenized hippocampi protein samples were resolved via 10% sodium dodecyl sulfate–polyacrylamide gel electrophoresis (SDS-PAGE) and subsequently transferred to polyvinylidene fluoride (PVDF) membranes (MilliporeSigma, Burlington, MA, USA). Following a blocking step with 5% bovine serum albumin (BioShop Canada Inc., Burlington, ON, Canada) in Tris-buffered saline containing 0.1% Tween-20 (TBST) for 1 h at ambient temperature, the membranes were incubated overnight at 4 °C with primary antibodies. The specific primary antibodies utilized included those targeting Aβ (cat. no. sc-28365; Santa Cruz Biotechnology, Inc., Dallas, TX, USA), p-Akt (cat. no. 4060; Cell Signaling Technology, Danvers, MA, USA), Akt (cat. no. 9272; Cell Signaling Technology, Inc.), IDE (cat. no. ab133561; Abcam), HSP70 (cat. no. 4872; Cell Signaling Technology, Inc.), and β-actin (cat. no. GTX110564; GeneTex, Irvine, CA, USA), with β-actin serving as the loading control for protein normalization. After washing with TBST, the membranes were incubated with the horseradish peroxidase (HRP)-conjugated secondary antibodies, goat anti-rabbit IgG (HRP) (cat. no. 111-035-003; Jackson ImmunoResearch Laboratories, Inc., West Grove, PA, USA) or goat anti-mouse IgG (HRP) (cat. no. GTX213111-01; GeneTex, Inc.). The dilutions of all primary and secondary antibodies were prepared in accordance with the manufacturer’s instructions. The membranes were visualized with an enhanced chemiluminescence substrate (Advansta, San Jose, CA, USA) and detected using the Amersham Imager 600 imaging system (AI600; GE Healthcare Life Sciences, Marlborough, MA, USA). The images were analyzed with Image Lab software (version 6.0.1, Bio-Rad Laboratories, Inc., Hercules, CA, USA).

### 4.11. Statistical Analysis

For statistical analysis, results were presented as mean ± standard error of the mean (SEM) and were evaluated using one-way analysis of variance (ANOVA) followed by Tukey’s post hoc test, conducted with OriginPro 2022 software (version 2022; OriginLab Corporation, Northampton, MA, USA). The *p*-value of less than 0.05 was considered statistically significant.

## 5. Conclusions

This study demonstrates that TCSNI may represent a safe and potentially effective approach to induce mild stress in vivo, with the aim of alleviating cognitive deficits associated with AD. TCSNI preserved olfactory function while significantly improving memory performance in Aβ-injected mice, as evidenced by Y-maze and novel object recognition tests. At the molecular level, TCSNI was associated with reduced hippocampal Aβ accumulation and increased expression of proteins implicated in Aβ clearance and neuronal survival, including HSP70, IDE, and p-Akt. These findings suggest that TCSNI could activate endogenous cellular protective mechanisms, potentially contributing to improved protein homeostasis and cognitive function. While promising, further studies are needed to explore additional stress-responsive pathways, evaluate the sustainability of the observed cognitive improvements, and validate these effects across different models and species, as well as to investigate the potential translation of TCSNI to clinical applications in humans.

## Figures and Tables

**Figure 1 ijms-26-10236-f001:**
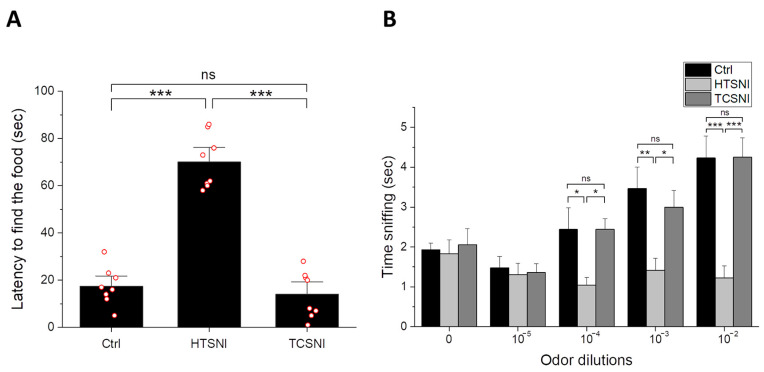
Effects of HTSNI and TCSNI on olfactory detection in mice. Olfactory performance was assessed using the buried food test and the olfactory habituation test, as described in Section 4: (**A**) Buried food test: latency to locate the buried food. (**B**) Olfactory sensitivity test: response time to odor cues. Data are presented as mean ± standard error of the mean. Each red circle represents one mouse. Statistical significance was determined using one-way analysis of variance (ANOVA) followed by Tukey’s post hoc test. (*n* = 8 in each group). Significance levels between the indicated groups are denoted as * *p* < 0.05, ** *p* < 0.01, *** *p* < 0.001, while non-significant differences are labeled as ns.

**Figure 2 ijms-26-10236-f002:**
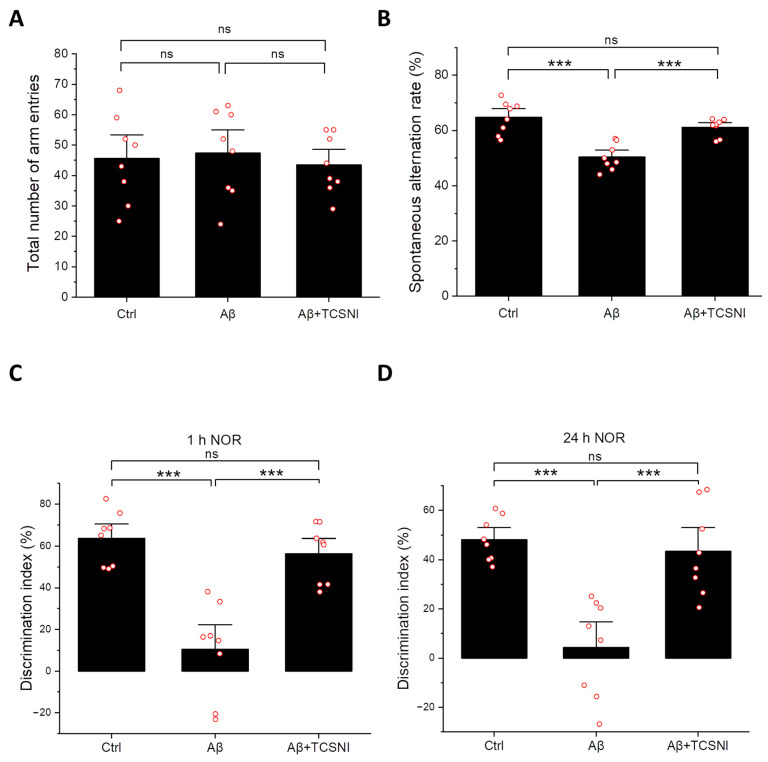
Effect of TCSNI applications on mice’s behavioral performance. The recovery effects of TCSNI treatment on Aβ-induced recognition impairments were assessed through (**A**) the total number of arm entries and (**B**) the spontaneous alternation index in the Y-maze test, and (**C**) the discrimination index with intertrial intervals of 1 h and (**D**) 24 h in the NOR maze test. Data are presented as mean ± standard error of the mean. Each red circle represents one mouse. Statistical significance was determined using one-way analysis of variance (ANOVA) followed by Tukey’s post hoc test. (*n* = 8 in each group). Significance levels between the indicated groups are denoted as *** *p* < 0.001, while non-significant differences are labeled as ns.

**Figure 3 ijms-26-10236-f003:**
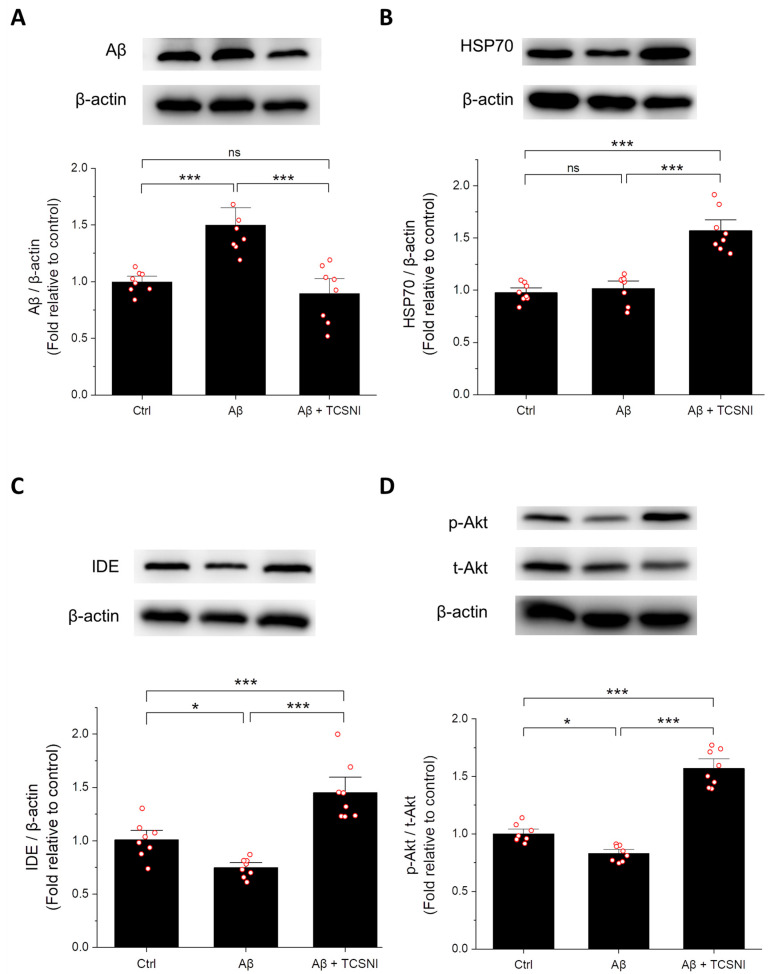
Effect of TCSNI applications on the expression levels of Aβ, HSP70, IDE, and p-Akt in the hippocampus. Representative Western blots and the quantifications of (**A**) Aβ, (**B**) HSP70, (**C**) IDE, and (**D**) p-Akt are shown to assess the anti-Aβ effect in the hippocampus. The protein levels of Aβ, HSP70, and IDE were normalized to β-actin, while p-Akt was normalized to t-Akt to accurately reflect the phosphorylation level relative to total Akt expression. Each relative expression level was compared to the control and expressed as a fold change relative to the control. Data are presented as mean ± standard error of the mean. Each red circle represents one mouse. Statistical significance was determined using one-way ANOVA followed by Tukey’s post hoc test. (*n* = 8 in each group). Significance levels between the indicated groups are denoted as * *p* < 0.05 and *** *p* < 0.001, while non-significant differences are labeled as ns.

**Figure 4 ijms-26-10236-f004:**
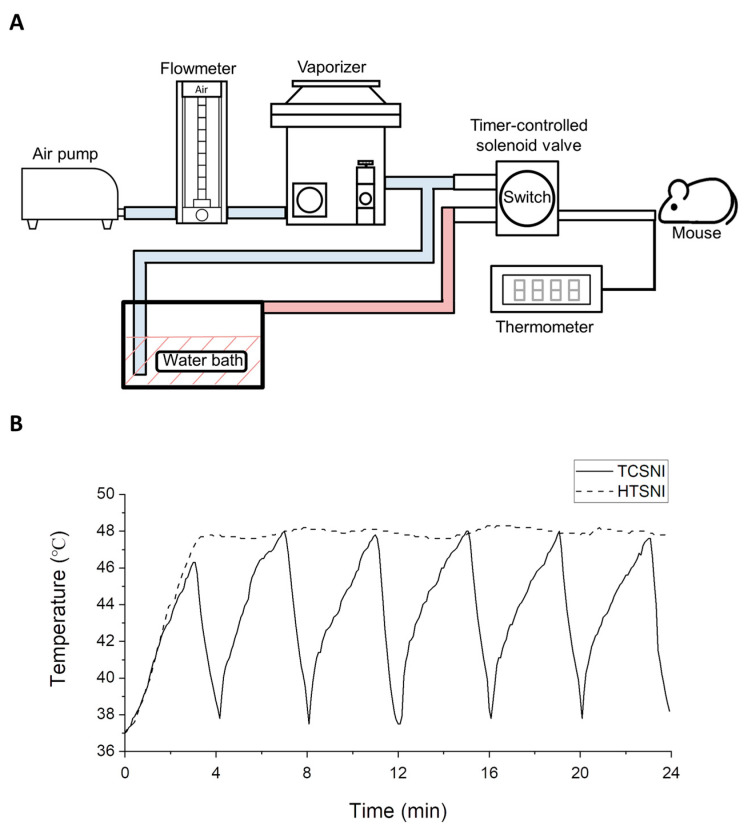
Experimental design and setup of TCSNI: (**A**) The TCSNI system is depicted, which provides inhaling mildly heated air for cyclic thermal stimulation. (**B**) Actual air temperatures entering the mouse’s nasal cavity were recorded using a K-type needle thermocouple throughout the exposure durations of TCSNI and HTSNI treatments.

**Figure 5 ijms-26-10236-f005:**
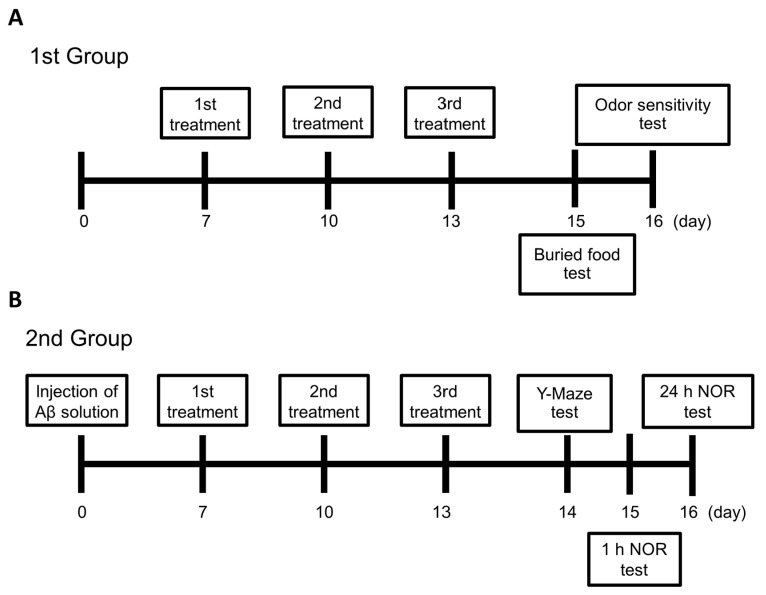
TCSNI treatment and assessment of animal behavior: (**A**) In the 1st group, the thermal applications were administered to the mice in the HTSNI and TCSNI groups on days 7, 10, and 13. The buried food and odor sensitivity tests were conducted on days 15 and 16, respectively. (**B**) In the 2nd group, Aβ solution was i.c.v.-injected into the mice’s brains of the Aβ and Aβ + TCSNI groups at day 0, and TCSNI application was performed on days 7, 10, and 13. The Y-maze test was conducted on day 14, followed by the NOR tests on days 15 and 16, respectively. After the NOR test, the mice were weighed and subsequently euthanized for brain tissue collection.

## Data Availability

The data presented in this study are available upon request from the corresponding author.

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
