# Peer review of "Thermal Cycling Stimulation via Nasal Inhalation Attenuates Aβ25–35-Induced Cognitive Deficits in C57BL/6 Mice"

_ijms, 2025, doi:10.3390/ijms262010236_

Round 1
Reviewer 1 Report
Comments and Suggestions for Authors
In the article "Thermal cyclic stimulation by nasal inhalation attenuates Aß25-35-induced cognitive deficit in C57BL/6 mice", the authors presented data on nasal inhalation causing thermocyclic hyperthermia (TCHT) in the development of thermocyclic stimulation providing cyclic stimulation to maintain the N-2-B pathway. In general, the idea of this work seems to be quite promising and interesting, since it bypasses the BBB, however, this method needs to be studied in more depth.
1. The Introduction section describes this problem quite informatively, however, this section needs to be edited and repetitions removed.
2. The Results section fully reflects the presented material and does not cause critical comments.
3. The Discussion section is written quite convincingly, but needs to be more clearly structured, in particular, the authors can highlight thematic sections and make this section of the article clearer.
4. Perhaps the authors should make a "Conclusion" section in this article, in which the main conclusions of the work are presented quite clearly and concisely.
Comments on the Quality of English Language
English needs editing
Author Response
Dear Reviewer 1,
We sincerely appreciate your thorough review of our manuscript and your recognition of the significance of our research. In response to your valuable comments and suggestions, we have revised the manuscript to improve its clarity and overall quality.
The following is a point-by-point response to your comments.
Comment 1:
The Introduction section describes this problem quite informatively, however, this section needs to be edited and repetitions removed.
Response 1:
We appreciate this constructive suggestion. The Introduction section has been carefully revised to remove redundant statements and improve readability. The corresponding revisions can be found on page 2 (lines 63–64, 69–71) and page 3 (lines 94–99) of the revised manuscript.
Comment 2:
The Results section fully reflects the presented material and does not cause critical comments.
Response 2:
We sincerely thank the reviewer for the positive comment and are pleased that the Results section was found to be clear and satisfactory.
Comment 3:
The Discussion section is written quite convincingly, but needs to be more clearly structured, in particular, the authors can highlight thematic sections and make this section of the article clearer.
Response 3:
We thank the reviewer for the valuable suggestion regarding the clarity and structure of the Discussion section. The Discussion has been revised so that each paragraph now clearly highlights its main theme, thereby improving the overall clarity. The corresponding revisions can be found on page 8 (lines 262–263, 272–274), page 9 (lines 299–301), and page 10 (lines 336–338, 350–354, 357–360) of the revised manuscript.
Comment 4:
Perhaps the authors should make a "Conclusion" section in this article, in which the main conclusions of the work are presented quite clearly and concisely.
Response 4:
We appreciate the reviewer’s helpful suggestion. A dedicated “Conclusion” section has been added to clearly and concisely summarize the main findings of our study. This revision can be found on pages 15–16 (lines 524–537) of the revised manuscript.
Finally, we would like to thank you again for your valuable comments. We believe that these revisions have significantly enhanced the manuscript and improved its overall quality.

Reviewer 2 Report
Comments and Suggestions for Authors
The authors of the manuscript studied a novel non-invasive intervention therapeutic strategy for Alzheimer’s disease (AD) treatment, i.e. inhalation of mildly heated air as thermal stimulation to induce mild stress and stimulate cerebral activity through the nose-to-brain (N-2-B) pathway. In behavioral tests (i.e. Y-maze and novel object recognition assessments), significant cognitive improvement was observed in β-amyloid (Aβ)-treated mice after thermal cycling-stimulation via nasal inhalation (TCSNI). The authors also demonstrated that the TCSNI approach down-regulated Aβ accumulation and elevated HSP70, IDE, and p-Akt levels in hippocampus areas of the mice, which is associated with the beneficial effects. The study contributes new knowledge on alternative therapeutic strategies for AD and establishment of TCSNI as a potential non-invasive treatment. This reviewer would like to recommend its publication after addressing the following questions/comments:
- In line 224 (Figure 3): Why was the protein level p-Akt normalized to t-Akt instead of β-actin. Explain how the data Figure 3D was obtained further.
- Clearly describe the composition of the gas used in the HTSNI approach. It seems to also include 3% isoflurane anesthetic gas used to sedate mice during the procedure. Could isoflurane be a factor to impact on the variation of protein levels in brain? In particular, Aβ accumulation was not included in control group.
- Discuss the duration of the treatment effects. How long had the improved cognitive behaviors persisted in mice, after just three times of the TCSNI treatments.
- Discuss how AD patients could benefit from the TCSNI treatments for long-term effects.
Author Response
Dear Reviewer 2,
We sincerely thank the reviewer for the careful evaluation of our manuscript and for providing constructive and insightful comments. We appreciate the reviewer’s recognition of the potential of our study in developing a novel non-invasive strategy for Alzheimer’s disease. In response to the reviewer’s suggestions, we have revised the manuscript and provided detailed explanations below.
The following is a point-by-point response to your comments.
Comment 1:
In line 224 (Figure 3): Why was the protein level p-Akt normalized to t-Akt instead of β-actin. Explain how the data Figure 3D was obtained further.
Response 1:
We thank the reviewer for this insightful comment. In this study, the phosphorylation level of Akt (p-Akt) was normalized to its total protein level (t-Akt) to accurately reflect the activation ratio, while β-actin was used as a loading control to confirm equal protein loading across samples. A brief explanation has been added to the Results section and the legend of Figure 3.
This revision can be found on page 6 (lines 208–209), and page 7 (lines 217–218) of the revised manuscript.
Comment 2:
Clearly describe the composition of the gas used in the HTSNI approach. It seems to also include 3% isoflurane anesthetic gas used to sedate mice during the procedure. Could isoflurane be a factor to impact on the variation of protein levels in brain? In particular, Aβ accumulation was not included in control group.
Response 2:
We appreciate the reviewer’s careful consideration of our manuscript. In the study, all experimental groups were subjected to the same isoflurane anesthesia during procedures. In addition, prior reports in various AD mouse models suggest that short-term isoflurane exposure is generally well tolerated and does not alter Aβ-related pathology. A brief note regarding anesthesia has been added to the Methods section to clarify this point. This revision can be found on page 11(lines 392-395) of the revised manuscript.
Comment 3:
Discuss the duration of the treatment effects. How long had the improved cognitive behaviors persisted in mice, after just three times of the TCSNI treatments.
Response 3:
We thank the reviewer for this valuable suggestion. Based on our current data, we acknowledge that only the short-term effects of TCSNI treatment on cognitive behaviors in mice can be observed. The persistence of these effects beyond the observed period remains to be investigated. We agree that evaluating the long-term sustainability of the observed cognitive improvements is an important direction for future studies. A brief discussion of this point has been added to the Discussion and Conclusion sections. This revision can be found on page 10 (lines 350–352) and pages 15–16 (lines 533–536) of the revised manuscript.
Comment 4:
Discuss how AD patients could benefit from the TCSNI treatments for long-term effects
Response 4:
We thank the reviewer for this valuable comment. TCSNI is a non-invasive and non-pharmacological approach with a straightforward application concept, making it a promising candidate for future therapeutic use in AD patients. At the same time, we acknowledge that further studies are required to evaluate its long-term effects. This perspective has been included in the Discussion and Conclusion sections to emphasize both the potential of TCSNI and the need for future investigation.This revision can be found on page 10 (lines 357–360) and page 16 (lines 536–537) of the revised manuscript.
Finally, we would like to thank the reviewer again for the valuable comments. We believe that these revisions have helped to improve the clarity and presentation of the manuscript.

Reviewer 3 Report
Comments and Suggestions for Authors
In the paper "Thermal cycling stimulation via nasal inhalation attenuates Aβ25-35-induced cognitive deficits in C57BL/6 mice", the authors present a novel proposal to stimulate brain activity through the N2B pathway minimizing thermal injuries.
The paper is well-written and allows for a fluid reading.
However, although the methodology used allows us to reach the conclusions shown by the authors, it would be interesting to compare the effects of treatment with TCSNI and HTSNI on Aβ and IDE, HSP70, and Akt expression levels in mice injected with Aβ25-35 under the same conditions. Although the authors report previous studies comparing HT versus TC-HT; and, in the present study, they report that HTSNI impaired olfactory function, while TCSNI did not show adverse effects, some different characteristics could be found. It is suggested to expand the discussion on this matter.
Likewise, since the researchers had the brains of the mice, it would have been interesting to evaluate the same markers in other areas of the brain. It is suggested to expand the discussion on this matter.
Author Response
Dear Reviewer 3,
We sincerely thank the reviewer for the thoughtful comments and positive feedback regarding our study. We are glad that the reviewer found the manuscript well-written and appreciated our novel approach using the N2B pathway to minimize thermal injuries.
The following is our response to your comments.
Comments and Suggestions for Authors
In the paper "Thermal cycling stimulation via nasal inhalation attenuates Aβ25-35-induced cognitive deficits in C57BL/6 mice", the authors present a novel proposal to stimulate brain activity through the N2B pathway minimizing thermal injuries.
The paper is well-written and allows for a fluid reading.
However, although the methodology used allows us to reach the conclusions shown by the authors, it would be interesting to compare the effects of treatment with TCSNI and HTSNI on Aβ and IDE, HSP70, and Akt expression levels in mice injected with Aβ25-35 under the same conditions. Although the authors report previous studies comparing HT versus TC-HT; and, in the present study, they report that HTSNI impaired olfactory function, while TCSNI did not show adverse effects, some different characteristics could be found. It is suggested to expand the discussion on this matter.
Likewise, since the researchers had the brains of the mice, it would have been interesting to evaluate the same markers in other areas of the brain. It is suggested to expand the discussion on this matter.
Response:
We sincerely thank the reviewer for the valuable comments. We note that this study represents the first attempt to use gas-mediated thermal stimulation to improve cognitive function in mice. Given that HTSNI may cause olfactory impairment, the present study focused on investigating the efficacy and underlying mechanisms of TCSNI, particularly regarding protein changes in the hippocampus associated with cognition. We agree that comparing TCSNI with HTSNI and evaluating molecular markers in other brain regions are of academic interest. These represent important directions for future studies, including comparisons of different stimulation modes and exploration in other species. Relevant discussion addressing these points has been incorporated into the Discussion and Conclusion sections of the revised manuscript.
This revision can be found on page 10 (lines 352–355) and pages 15–16 (lines 533–536) of the revised manuscript.
Finally, we sincerely thank the reviewer again for the valuable comments, which have helped us clarify and expand the discussion in the revised manuscript.
